# SARS-CoV-2 Infection and Anemia—A Focus on RBC Deformability and Membrane Proteomics—Integrated Observational Prospective Study

**DOI:** 10.3390/microorganisms12030453

**Published:** 2024-02-23

**Authors:** Angelo D’Alessandro, Elena Krisnevskaya, Valentina Leguizamon, Ines Hernández, Carolina de la Torre, Joan-Josep Bech, Josep-Tomàs Navarro, Joan-Lluis Vives-Corrons

**Affiliations:** 1Medical Campus, University of Colorado Anschutz, Aurora, CO 80045, USA; angelo.dalessandro@cuanschutz.edu; 2Red Blood Cells and Haematopoietic Disorders, Josep Carreras Institute for Leukaemia Research (IJC), 08916 Badalona, Spain; ekrish6@gmail.com (E.K.); valen.velilla.vl@gmail.com (V.L.); 3Josep Carreras Leukaemia Research Institute, Haematology Department, ICO-Germans Trias i Pujol Hospital, Autonomous University of Barcelona, 08916 Badalona, Spain; ihernandez@iconcologia.net (I.H.); tnavarro@carrerasresearch.org (J.-T.N.); 4Proteomic Unit at Josep Carreras Leukaemia Research Institute IJC Building, Campus ICO-Germans Trias i Pujol, 08916 Badalona, Spain; cdelatorre@carrerasresearch.org (C.d.l.T.); jbech@carrerasresearch.org (J.-J.B.)

**Keywords:** COVID-19, anemia, red blood cells, membranopathies, enzymopathies, ektacytometry, proteomics

## Abstract

Introduction: The multifaceted impact of COVID-19 extends beyond the respiratory system, encompassing intricate interactions with various physiological systems. This study elucidates the potential association between SARS-CoV-2 infection and anemia, with a particular emphasis on the deformability of red blood cells (RBCs), stability of hemoglobin, enzymatic activities, and proteomic profiles. Methods: The study encompasses a cohort of 74 individuals, including individuals positive for COVID-19, a control group, and patients with other viral infections to discern the specific effects attributable to COVID-19. The analysis of red blood cells was focused on deformability measured by osmotic gradient ektacytometry, hemoglobin stability, and glycolytic enzyme activity. Furthermore, membrane proteins were examined using advanced proteomics techniques to capture molecular-level changes. Results: Findings from the study suggest a correlation between anemia and exacerbated outcomes in COVID-19 patients, marked by significant elevations in d-dimer, serum procalcitonin, creatinine, and blood urea nitrogen (BUN) levels. These observations suggest that chronic kidney disease (CKD) may play a role in the development of anemia in COVID-19 patients, particularly those of advanced age with comorbidities. Furthermore, the proteomic analyses have highlighted a complex relationship between omics data and RBC parameters, enriching our understanding of the mechanisms underlying the disease. Conclusions: This research substantiates the complex interrelationship between COVID-19 and anemia, with a specific emphasis on the potential repercussions of SARS-CoV-2 infection on RBCs. The findings contribute to the growing body of evidence supporting the extensive impact of COVID-19 on RBCs.

## 1. Introduction

The coronavirus infection (COVID-19) is caused by the SARS-CoV-2 virus and manifests primarily through symptoms such as shortness of breath, persistent cough, and fever. Given the critical role of red blood cells (RBCs) in oxygen transport and gas exchange, understanding the impact of COVID-19 on RBC properties is imperative. Previous studies have demonstrated alterations in the structural and functional proteins of RBCs in patients with COVID-19, correlated with disease severity and inflammatory markers such as interleukin-6 [1]. These alterations include the cleavage of the N-terminus cytosolic domain of band 3 (SLC4A1), attributable to oxidant stress or proteolytic activity [2] and changes in the band 3 interactome [3] involving key structural proteins such as ankyrin (ANK1) and spectrin (SPTA1 and SPTB). Subsequent research has further identified morphological changes in RBCs from COVID-19 patients and correlated these with proteome functional alterations [4,5]. While the mechanistic link to an increased hemolytic propensity of RBCs remains to be fully elucidated, a significant correlation has been established between altered red cell distribution widths (RDWs) and disease severity, indicating RDW as a potential marker for clinical outcomes in COVID-19 [6,7,8]. Additionally, Bergamaschi G et al. [9] have found in COVID-19-positive patients, the coexistence of anemia in 61% of cases compared to 45% of cases in a group of patients with clinical and laboratory findings suggestive of COVID-19, but with negative nasopharyngeal swab test. Recently, a comprehensive meta-analysis has highlighted a progressive decrease in hemoglobin levels as indicative of worse clinical progression in COVID-19 patients, pointing towards the potential role of SARS-CoV-2 RNA in contributing to anemia through effects on RBC structure [10].

It is well established that RBCs can be targeted by pathogens [11], leading to direct intravascular hemolysis or indirect clearance by reticuloendothelial systems. Given that RBCs cannot support viral replication, the possibility of SARS-CoV-2 RNA invasion into RBCs, similar to the behavior of flaviviruses like Zika, raises important questions regarding the unique or universal nature of RBC alterations in response to infections. In this context, it is worth mentioning that the activation of cGAS-STING-interferon-IDO1-kynurenine responses is a key determinant of prognosis in COVID-19 patients [12,13,14], yet these responses are universally triggered by almost any infective pathogen [15], especially when a background chronic inflammatory condition in aging is present. 

Several years ago, an inhibition of liver pyruvate kinase (PKL) by physical interaction between the SARS-CoV-2 nucleocapsid protein and the enzymatic protein was described [16]. Since PKL and RBC PK (PKLR) share genetic encoding, a similar mechanism for reduced RBC lifespan and PKLR activity in COVID-19 in patients with COVID-19 infection was suggested. Unfortunately, this hypothesis could not be demonstrated in the present study. 

Despite the frequent occurrence of anemia in COVID-19 patients, the direct link between the infection and anemia remains to be conclusively demonstrated, particularly once common causes related to iron metabolism and inflammatory conditions are excluded. This study aims to investigate the effects of COVID-19 on RBC (Hb), enzyme activities, and deformability in an effort to elucidate the mechanisms by which SARS-CoV-2 RNA may influence RBC dysfunction and contribute to anemia. Interestingly, a prevalence of anemia in COVID-19 patients has been documented, with a notable difference in incidence between patients testing positive for SARS-CoV-2 and those with similar symptoms but negative test results. The examination of proteomic biomarkers has contributed to enhancing the prediction of severe disease outcomes to improve the understanding of viral mechanisms and to explore more effective treatment strategies.

## 2. Materials and Methods

### 2.1. Patient Enrollment

The experimental design was based on an observational prospective study of a single cohort of 74 individuals (63 patients and 11 controls). Inclusion criteria were age >18 years and documented diagnosis of SARS-CoV-2 infection, confirmed by RT-PCR performed in at least one nasal/pharyngeal swab specimen. Viral infections with similar clinical COVID-19+ phenotype and negative RT-PCR in a nasal/pharyngeal swab specimen were also included. Exclusion criteria were age <18 years, not capable of signing the Informed Consent (IC), and a known history of a hereditary RBC defect. Accordingly, on the basis of anemia and the RT-PCR positivity, the patients were classified into four groups where patients negative for RT-PCR had viral infections not due to COVID-19 (Table 1). A fifth group with 11 healthy blood donors was included as the control group. Anemia was defined according to WHO (<120 g/L for women and <130 g/L for men). For the classification of the disease severity, the recommendations of the European Centre for Disease Prevention and Control [17] and the National Health Commission of the People’s Republic of China and WHO [18] were used. Accordingly, patients with COVID-19 infection were classified into 4 categories: 1. Mild clinical symptoms without pneumonia at chest computed tomography; 2. Moderate fever and other respiratory symptoms with pneumonia seen at imaging; 3. Severe respiratory distress (≥30 breaths per min), hypoxia (oxygen saturation: ≤93%), or abnormal results of blood gas analysis; and 4. Critical respiratory failure requiring mechanical ventilation, shock, or other organ(s) failure requiring intensive care unit monitoring and treatment. It should be mentioned that the study was performed during the pandemic and immediately after admission. For this reason, the main treatments received and the patient’s clinical follow-up have not been considered here. The study with Ref. CEI PI-21-30-09 was approved by the Hospital Human Ethics Committee, and Informed Consent was signed by all the patients included in the study and performed in respect of the Declaration of Helsinki.

### 2.2. Hematological and Biochemical Parameters

Complete Blood Count (CBC) and basic biochemistry parameters were tested in all patients with SARS-CoV-2 positivity and other viral infections, as well as in the control group. In all patients with anemia, iron study, vitamin B12 and folate, C-reactive protein (CRP), and hepcidin were also tested. In all patients, hemoglobin stability was measured using the isopropanol test [19], and RBC enzyme activities were measured according to Beutler [20] with slight modifications [19].

### 2.3. RBC Deformability

RBC deformability was determined with a new-generation ektacytometer—a laser diffractometer that measures the deformability of an RBC population exposed to an increasing osmotic gradient under a constant shear stress [21]. The osmotic gradient ektacytometry (OGE) measures RBC geometry, cytoplasm viscosity, cell volume, and membrane fluidity by the RBCs’ shift from discoid to elliptical shape, gauged by light scatters as the cell responds to shear forces over an osmotic gradient. In addition to rheological parameters, the ektacytometer allows for identification of eventual RBC structural abnormalities affecting membrane or hemoglobin content. The OGE profile obtained with the osmoscan module is a characteristic curve that shows the amount of deformability on the *y*-axis and the osmolality on the *x*-axis (Figure 1), as well as three main parameters: (a) EImax/Omax or the value of osmolality at maximum EI (deformability), (b) EImin/Omin, or the value at which RBCs have attained their critical hemolytic value due to osmotic shifting of water into the cell in a hypotonic environment (osmotic fragility), and (c) EIhyper/Ohyper, or the osmolality at which the index is midway between the maximal deformability and Omin (cellular hydration).

### 2.4. RBC Membrane Proteins (Proteomics)

Of the total cohort of 74 patients, RBC samples were available from 32 subjects for proteomic analyses. Specifically, proteomic analyses were performed on 11 healthy control subjects, patients with COVID-19 (with and without anemia), and patients with other viral infections (with and without anemia) (*n* = 6 for each one of these four groups). Patient samples were collected by centrifugation for 5 min at 150× *g* and were suspended in PBS containing 1 mM EDTA. Then, the slurry was passed through a leukocyte depletion filter. RBCs were washed four times with PBS and stored overnight on ice at 4 °C in PBS containing 10 mM glucose and EDTA-free protease inhibitor cocktail tablets (Roche, Mannheim, Germany). Proteomic analysis was performed on the day of collection, and the steps used for sample treatment are described below.

Sample preparation: RBCs were isolated according to Pesciotta et al. [22] and lysed in 0.1 M Tris-HCl containing 2% SDS and 0.05 M DTT at 100 °C for 5 min. RBC membranes were isolated by centrifugation at 21,000× *g* for 40 min and purified by 4–5 additional washes and centrifugations. Protein was quantified with the DC™ Protein Assay Kit II (Bio-Rad, Hercules, CA, USA).

Preparation of samples for protein digestion and peptide purification prior to mass spectrometry analysis was performed according to the conventional procedures for cell lysis, protein precipitation, and protein quantification. Prior to digestion, samples were reduced and alkylated with DTT and CAA; then, they were diluted with Tris 0.1 M to reach urea 2 mol/L. Lys-C was added at 1:100 (*w*/*w*) (enzyme-to-protein ratio), and protein digestion was carried out at 30 °C ON (overnight). The samples were diluted with Tris buffer (0.1 M) to achieve a final urea concentration of 0.8 mol/L. Trypsin was added to the diluted samples at a ratio of 1:100 (enzyme-to-protein ratio), and protein digestion was performed again at 30 °C for a duration of 8 h, allowing the trypsin enzyme to cleave the proteins into smaller peptides. The enzymatic reaction was stopped by adding formic acid (FA) to a final concentration of 10% (*v*/*v*). Digested peptides were cleared by centrifugation and purified using a reversed-phase C18 Microspin column according to the manufacturer’s instructions. Elution of peptides was performed with 50% ACN in 0.1% TFA; then, peptides were dried by the speedVac at RT and stored at −80 °C until further processing. 

LC-MS/MS Measurements: Tryptic peptide samples were reconstituted with 3% ACN and 0.1% FA aqueous solution at 100 ng/µL, and 8 µL (800 ng) was loaded into the Evotip. Peptides were separated using an Evosep EV1106 column (150 μm × 150 mm, 1.9 μm) (Evosep, Odense, Denmark) at a flow rate of 500 nL/min with an 88 min run. The column outlet was directly connected to an EASY-Spray source (Thermo Fisher Scientific, Waltham, MA, USA) fitted on an Orbitrap Eclipse™ Tribrid Mass Spectrometer (Thermo Fisher Scientific). The mass spectrometer was operated in a data-dependent acquisition (DDA) mode. In each data collection cycle, one full MS scan (375–1500 *m*/*z*) was acquired in the Orbitrap (1.2 × 105 resolution setting and automatic gain control (AGC) of 2 × 105). Ions were fragmented in the HCD with a collision energy of 28%, 0.25 activation Q, an AGC target of 5 × 104, an isolation window of 0.7 Da, a maximum ion accumulation time of 54 ms, and a turbo ion scan rate. Previously analyzed precursor ions were dynamically excluded for 15 s. MS2 were detected in the LC-MS Orbitrap (Thermo Fisher, Waltham, MA, USA) with a resolution of 30,000.

Data analysis and statistics for proteomic analyses: The .RAW files were processed using the MaxQuant 2.1.3.0 software. The peak lists were searched against a SwissProt Human Database (https://www.uniprot.org/ downloaded in 3 March 2021) with the help of the MaxQuant built-in Andromeda: a peptide search engine integrated into the MaxQuant environment. The false discovery rate (FDR) was assessed by using a decoy database. Trypsin was selected as the enzyme, and a maximum of two missed cleavages were allowed. Carbamidomethylating in cysteines was set as a fixed modification, whereas oxidation in methionine, acetylation at the protein N-terminal, and deamidation at asparagines and glutamines were used as variable modifications. Searches were performed using a peptide tolerance of 7 ppm and a product ion tolerance of 0.5 Da. The resulting data files were filtered for FDR. The quantification was performed using the iBAQ algorithm. The list of proteins was filtered to remove the ‘potential contaminants’, ‘reverse’, and ‘only identified by site’ proteins. The iBAQ-based intensity values were log2 transformed and analyzed to look for potential outliers, which were removed from the final data matrix. Additionally, we removed from the dataset the samples showing significant non-specific deformability alterations (ektacytometry). The samples present in the final data matrix were then normalized by median-centering the iBAQ intensity values. The statistical analysis was performed with the help of R (https://cran.r-project.org/) and Rstudio (https://www.rstudio.com) using the ‘limma’ package.

## 3. Results

### 3.1. General Hematological Data, Iron Metabolism and Inflammation Parameters

Table 2 summarizes the main demographic and laboratory parameters studied in the patients with laboratory-confirmed COVID-19+ compared to patients with viral infection but negative nasopharyngeal swab test and the control group. At hospital admission, the 20 patients of Group 2 (COVID-19+ with anemia) exhibited a more severe disease than the 13 patients of Group 1 (COVID-19+ without anemia). This difference was not observed between the 20 patients of Group 4 (viral infection with anemia) and the 10 patients of Group 3 (viral infection without anemia). WBC count was only slightly increased in patients COVID+ with anemia but moderately increased in patients with viral infection (with or without anemia). Platelet count was normal in all the patient groups. No association was found between hemoglobin concentration and the neutrophil-to-lymphocyte or the platelet-to-lymphocyte ratios. The d-dimer, serum procalcitonin, creatinine, and BUN were significantly increased in all the patients studied here, but especially in COVID-19+ patients with anemia (Group 2). Alkaline phosphatase, PCR, and hepcidin were also significantly increased in all groups, but especially in patients with viral infection and anemia (Group 4). Finally, iron metabolism parameters (serum iron, serum ferritin, and transferrin saturation index) were markedly abnormal in the same Group 4 when compared to the other groups. 

### 3.2. Hemoglobin Stability

The isopropanol test was normal in all the patients included in this study, with the exception of five viral-infected patients—three with anemia and two without anemia. These patients were not the same as those with altered left-shifted curves. The study of these patients with high-performance liquid chromatography (HPLC) discarded unstable hemoglobinopathies.

### 3.3. RBC Enzyme Activities

RBC enzyme activities from the glycolytic pathway and the oxidative metabolism measured in all patients are summarized in Table 3. All enzyme activities were found to be within normal range, with the exception of pyruvate kinase (PK) and adenylate kinase (AK). PK activity was significantly (*p* < 0.05) increased in COVID-19-positive patients with anemia (Group 2), and the AK activity was significantly (*p* < 0.05) increased in patients with anemia only (Groups 2 and 4). None of these increased enzyme activities were accompanied by an increased number of reticulocytes and/or circulating erythroblasts. No significant RBC morphological abnormalities were found.

### 3.4. RBC Deformability

All patients showed a normal OGE profile, with the exception of five patients with anemia—two COVID-19 positive and three COVID-19 negative—that exhibited a dehydrated RBC pattern of unknown origin. The genetic study of these patients by WES and a complete molecular panel for hemoglobinopathies (alpha and beta) discarded a hereditary RBC defect. The results of OGE parameters are summarized in Table 4. RBC deformability or maximum elongation index (EImax) was normal in all the patients, and only a significantly (*p* < 0.05) slight RBC overhydration (Ohyper and Area) was observed in Groups 2, 3, and 4. Usually, the normal range in Ohyper regions is rather wide, and the statistical significance obtained does not have great significance. 

### 3.5. RBC Membrane Proteins (Proteomics)

An overview of the experimental design for the proteomic analyses performed herein is provided in Figure 2A. Significant proteomic alterations were observed across the four groups of patients, with the most significant alterations in Group 2 (COVID-19+ with anemia). Such differences are highlighted by unsupervised hierarchical clustering analysis (HCA—Figure 2B) and partially (PLS-DA—Figure 2C,D). A list of the protein variables with the largest loading weights from this analysis is provided in Figure 2C. Of note, this list includes levels of transferrin (TF), as measured by proteomics, and discriminating between anemic and non-anemic patients. The list also included a series of structural proteins (especially myosin heavy chains MYH10) and enzymes involved in immune functions, such as myeloperoxidase (MPO) and lysozyme (LYZ), suggesting elevated levels of these proteins in the residual plasma or cellular fraction enriched for the study of RBCs, despite buffy coat depletion. While Group 2 patients (COVID-19+ with anemia) showed the highest degree of separation from the other groups across principal component 1 (PC1), patients with infection other than SARS-CoV-2 showed the highest degree of separation across PC2 compared to the rest of the tested samples (Figure 2D).

The patients from Group 3 (viral infection without anemia) were characterized by a significant alteration in a subset of the proteome (Figure 3A), with significant depletion of several RBC membrane proteins (SLC4A1, ANK1, SPTA1, SPTB, stomatin (STOM), band 4.1 E4BP1, flotillin (FLOT2), glycolytic enzymes (fructose bisphosphate aldolase (ALD), glyceraldehyde-3-phosphate dehydrogenase (GAPDH)) and hemoglobin/heme metabolism enzymes (HBA, HBB, and biliverdin reductase B–BLVRB) (Figure 3B). This group was also characterized by an elevation in transcription factors and autophagy-related proteins, with a notable elevation in amyloid protein (APP) and mitochondrial malate dehydrogenase 2 (MDH2), suggesting an elevation in nucleated and mitochondria-containing blood cells in the RBC fraction tested herein (e.g., reticulocytes) (Figure 3C). This may be the result of stress erythropoiesis, resulting in incomplete mitochondrial clearance during the final phases of maturation, as reported by multiple groups, including ours, in the context of sickle cell disease.

The patients from Group 4 (viral infection with anemia) showed a significant decrease in several proteins (Figure 4A), mostly involved in infection responses (top enriched pathway suggests depression in components of the immune cascades activated by bacterial—*Staphylococcus aureus*—and viral—human papillomavirus—infections (Figure 4B)), suggesting that stronger immune responses are elicited by SARS-CoV-2 independently from anemia, while such responses are blunted in patients suffering from other viral infections in the context of anemia. The combination of anemia and viral infection was associated with elevated levels of components of the vesiculation machinery (Figure 4C).

In the patients from Group 2 (COVID-19+ with anemia), the anemia had a strong effect that can be appreciated at a glance from unsupervised hierarchical cluster analysis (HCA) and volcano plot analyses (Figure 5A,B). In particular, anemia was associated with elevation in protein components involved in necroptosis and AMPK signaling, with corresponding decreases in the levels of proteins involved in MAPK signaling and autophagy (Figure 5C,D). Pathway analyses of significantly increased and decreased proteins in anemic patients compared to non-anemic ones indicate an enrichment in cGMP-PKG signaling and mitochondrial components that participate in oxidative phosphorylation events, suggesting an association between anemia and the increase in young RBCs containing mitochondria. On the other hand, decreasing proteins in anemic RBC fractions included components of the autophagy and mTOR pathways.

### 3.6. Omics Correlates to Clinical, Hematological, and Deformability Parameters

To determine the potential translational relevance of the proteomics observations, we correlated proteomics results to clinical, hematological, and biochemical parameters and RBC deformability measured by ektacytometry. Results showed four main clusters of highly connected variables. The largest cluster includes multiple hematological parameters (including RBC count, Hb concentration, and RDW), patient’s age, markers of renal function (BUN and creatinine), a series of RBC structural and membrane proteins (E4BP2, SPTA1, CD44, ANKFY1), hypoxia-upregulated 1 (HYOU1), and antioxidant capacity (reduced glutathione, glutathione peroxidase). A second cluster involved multiple markers of anemia (transferrin and serum iron), strongly associated with markers of one-carbon metabolism (B12, folate), liver metabolism (GGT, ALT, AST, LDH), ADP-ribosylation factors 1 and 3 (ARF1, ARF3), and the structural protein desmatin (DMTN). A third large cluster involved all parameters derived from the ektacytometric analysis, including the area of the osmoscan curve, and several proteins, including ACTBL2, 55kDa erythrocyte membrane protein (MPP1), and leucine-rich repeat-containing protein (LRRC57). Sub-networks included strongly connected hubs of transferrin and alfa-spectrin (SPTB), glycolytic and redox enzymes (HK, GSH), structural components (GYPC, FLOT2, ANK1), and RBC deformability (EImax). To further delve into these results, we plotted smile plots of Spearman correlates to patient age, RDW, PK activity, renal function (creatinine) and coagulability (d-dimer), common comorbidity factors in patients with severe COVID-19 infection (Figure 6). Top correlates to patient age indicate an age-dependent increase in renal dysfunction, which, in turn, is associated with markers of inflammation (e.g., RCP). Age and inflammation (IL-6, a marker of COVID-19 severity), as well as glycolytic enzymes (enolase–ENO1), also ranked amongst the top correlates to d-dimer measurements. As an internal validation of the quality of these correlative results, enzymatic assays of PK activity correlated significantly with the levels of RBC-PK isoforms (PKR) as detected via proteomics. The negative association between patient age and proteosome components (PSMD12, PSMA4) suggests a potential failure of protein degradation machinery in older subjects. Elevation in hemoglobin gamma chains (1 and 2) was negatively associated with PK activity and kidney function (elevated creatinine), suggestive of a linkage between anemia, kidney damage, and eventual subclinical hemolysis.

## 4. Discussion

The multifaceted etiology and prognosis of numerous clinical conditions, including those associated with respiratory complications such as COVID-19 infection, are significantly influenced by anemia [23,24]. Iron deficiency anemia (IDA) has been identified as a predisposing factor for lower respiratory tract infections in the pediatric demographic [25]. In adults, its presence upon hospital admission has been implicated as a potential determinant for adverse COVID-19 outcomes [26,27]. Recent investigations have linked anemia, particularly in the context of COVID-19, to increased mortality rates attributed to immune-mediated disruptions in iron homeostasis [28]. Furthermore, a reduction in hemoglobin levels has been observed in critically ill patients [29,30], underscoring the complex relationship between anemia and severe COVID-19 illness, which remains insufficiently elucidated. The pathogenesis of multifactorial anemia in SARS-CoV-2-infected individuals encompasses several mechanisms, including potential hemolysis induced by viral entry through erythrocyte membrane receptors [31], impaired erythropoiesis due to hematopoietic precursor invasion [32,33,34], and altered iron metabolism driven by pro-inflammatory cytokine-mediated upregulation of hepcidin [24,35]. However, since clinical and omics characterization of RBCs and plasma from COVID-19+ patients failed to document the increase in hemolysis, it can be assumed that the association of decreased iron availability, elevated levels of acute phase reactants, and a hindered erythropoiesis may explain the anemia of most COVID-19-infected patients [36].

This study introduces additional factors contributing to anemia in COVID-19 patients, emphasizing the comparative analysis of patients with other viral infections to enhance the understanding of this condition. Significantly, this research found elevated levels of d-dimer, procalcitonin (PCT), and markers of chronic kidney disease (CKD) in COVID-19 patients with anemia, suggesting a link between CKD and anemia in these patients. This association was further supported by a recent meta-analysis indicating a direct influence of aging and concurrent CKD on anemia in COVID-19 patients [37] in like fashion to patients suffering from CKD [38]. The study also highlights the role of elevated comorbidities, including renal dysfunction and alterations in erythrocyte structural membrane proteins, in the pathogenesis of anemia. Despite these findings, erythrocyte deformability was generally normal across the cohort, suggesting the complexity of anemia’s impact on patient outcomes. Increased serum ferritin levels in anemic COVID-19 patients were noted, indicating inflammation. However, similar observations in patients with other viral infections suggest a broader context of anemia beyond COVID-19. Altogether, these observations may contribute to explaining the role of anemia and hypoxia [39]. The study underscores the importance of timely anemia management in hospitalized COVID-19 patients, suggesting that oxygen supplementation or steroids, in addition to standard care, may mitigate deterioration [40]. In this view, it is worth highlighting the reassuring evidence on the potential impact of COVID-19 on RBC oxygen kinetics [41] despite evidence of a potential disruption in the so-called oxygen-dependent metabolic modulation revolving around band 3 stability [1]. Furthermore, the analysis of hemoglobin stability and erythrocyte enzyme activities, including pyruvate kinase (PK) and adenylate kinase (AK), offers insights into the metabolic alterations associated with COVID-19 and anemia and its probable contribution to the prognostic role of anemia in COVID-19 Patients [42] Concerning hemoglobin, the coronavirus, similar to other viruses, is able to interact with protoporphyrin IX through the spike protein involving beta chains of Hb [43], causing eventual Hb denaturation and the inhibition of viral replication by blocking the SARS-CoV-2-cell fusion mediated by the spike protein [44]. Using the isopropanol test, we have analyzed the stability of the hemoglobin molecule in all the patients included in this study, and with the exception of five cases, all the patients exhibited a normal hemoglobin stability. Probably, in mature RBCs, the interaction between SARS-CoV-2 and hemoglobin can take place, but since the virus replication is prevented by the absence of a nucleus, the final effect on hemoglobin stability is not significant. The same may happen at the bone marrow level, where the virus enters the nascent erythroblasts through CD147 and CD26 [31]. Here, even though the virus can replicate, there may be no significant effect on hemoglobin stability. Concerning RBC enzyme activities, their measurement was performed in all the patients included in this study and showed normal values, with the exception of pyruvate kinase (PK) and adenylate kinase (AK), which exhibited a significant (*p* < 0.05) increase in activity in both COVID-19+ and virally infected patients with anemia. Previous studies on RBCs from COVID-19 patients have highlighted an increase in the glycolytic pathway manifested by a characteristic increase in glucose consumption accompanied by an accumulation of intermediates of glycolysis and higher levels of phosphofructokinase (PFK), the rate-limiting enzyme of glycolysis. Mammals have two pyruvate kinase genes, PK-LR and PK-M. PK-LR encodes for two PK isozymes: PKL (liver) and PKR (RBCs). The PK-M gene encodes for pyruvate kinase isozyme M1 (muscle and brain) and M2 (leukocytes and early fetal tissues). Only PKLR encodes for the RBC isozyme, which is affected in PK deficiency [45,46]. However, it has been shown that patients with severe COVID-19 disease exhibit a higher expression of leukocyte PKM2, suggesting that increased PKM2 is involved in the metabolic reprogramming process participating in the immune response induced by COVID-19 [47]. Adenylate kinase (AK) is the key enzyme of nucleotide metabolism and belongs to the nucleoside monophosphate kinase (NMPK) family [48]. The mechanism/s of the increased AK (AK1) activity in COVID-19+ patients with anemia is unknown, but increased AK, together with other biomarkers, can be helpful in assessing the risk of diseases where oxidative/inflammatory stress plays a crucial role in pathogenesis [49,50]. Of note, here, AK1 levels positively correlated with increases in RDW, a marker of disease severity and prognosis in COVID-19 patients [6,51]. This research sheds light on the direct effects of SARS-CoV-2 on erythrocyte structural proteins and metabolic pathways, potentially contributing to thromboembolic and coagulopathic complications [52]. Finally, the minimal changes observed through ektacytometry, despite proteomic evidence, pose a conundrum. It is plausible that, within our cohort, the protein modifications induced by viral interaction were not substantial enough to significantly alter red blood cell deformability.

In conclusion, this study highlights the complexity of RBC dynamics in the context of viral infections and enhances the understanding of anemia as a significant factor in the severity of outcomes in SARS-CoV-2-infected patients. Unfortunately, although these compelling findings are encouraging, the study’s limitations, including a modest cohort size and geographical restrictions, suggest caution in generalizing the results, and we advocate for a deeper exploration of erythrocyte dynamics in viral infections.

## Figures and Tables

**Figure 1 microorganisms-12-00453-f001:**
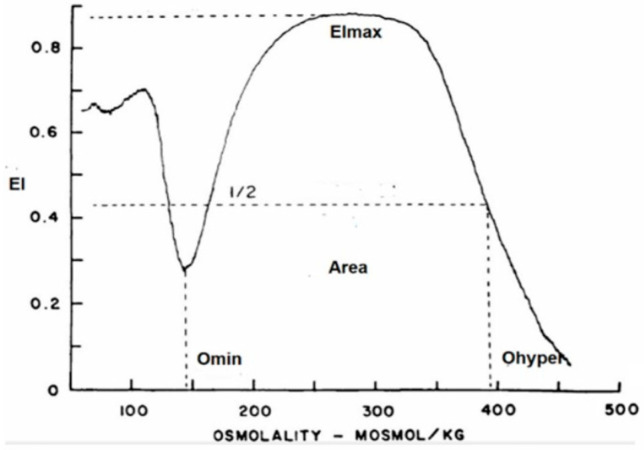
Graphic representation of the OGE profile obtained with the osmoscan module of the LoRRca ektacytometer. *y*-axis: elongation index (EI); *x*-axis: osmolality (mOsmol/Kg).

**Figure 2 microorganisms-12-00453-f002:**
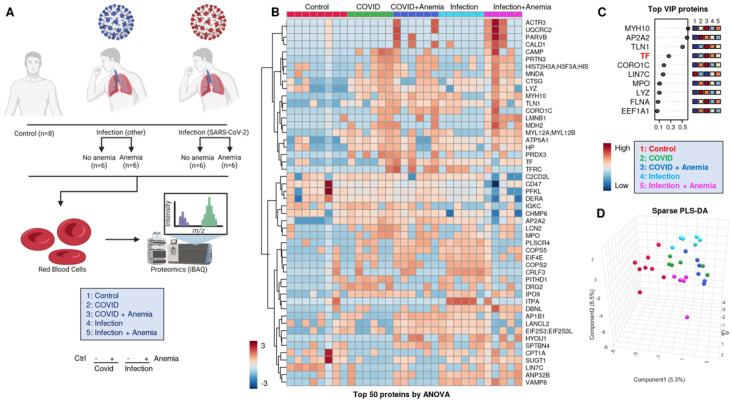
(**A**) Overview of the experimental design for the proteomic analyses performed here. (**B**) Proteomics alterations observed across the four groups of patients, with the differences highlighted by unsupervised hierarchical clustering analysis (HCA) and (**C**) with partially supervised partial least square-discriminant analysis (PLS-DA), including (**D**) a list of the protein variables with the largest loading weights.

**Figure 3 microorganisms-12-00453-f003:**
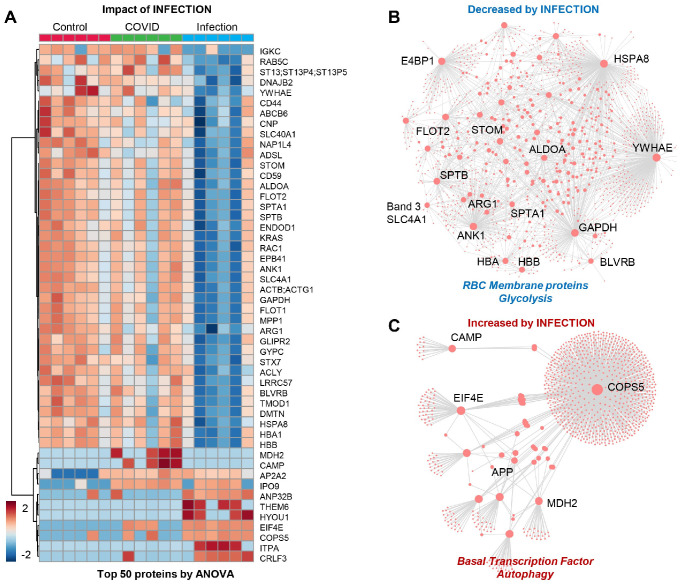
(Patients of Group 3) (**A**) Impact of infection on a subset of the proteome with: (**B**) significant depletion of several RBC membrane proteins and glycolytic enzymes and hemoglobin/heme metabolism and (**C**) elevation in transcription factors and autophagy-related proteins, especially amyloid protein (APP) and mitochondrial malate dehydrogenase 2 (MDH2).

**Figure 4 microorganisms-12-00453-f004:**
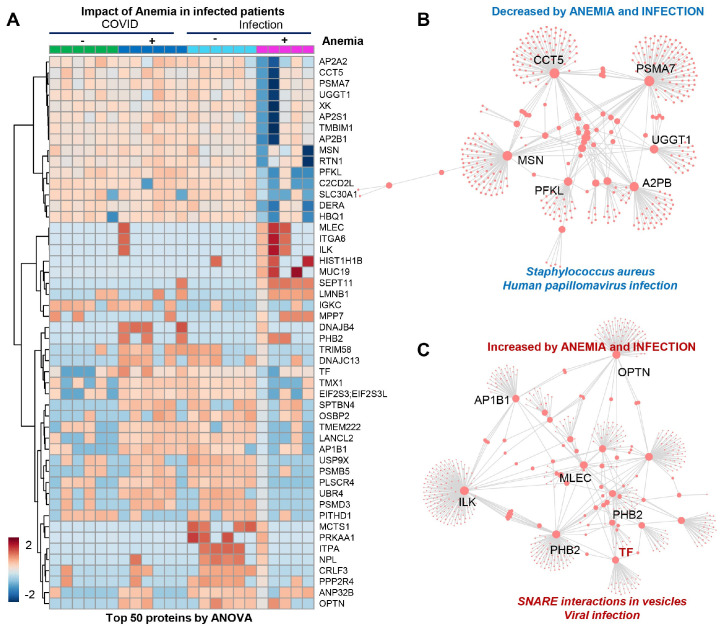
(Patients of Group 4) (**A**) Significant decrease in several proteins, mostly involved in infection responses. (**B**) Depression in components of the immune cascades activated by bacterial *Staphylococcus aureus* and viral infections. (**C**) Elevated levels of components of the vesiculation machinery.

**Figure 5 microorganisms-12-00453-f005:**
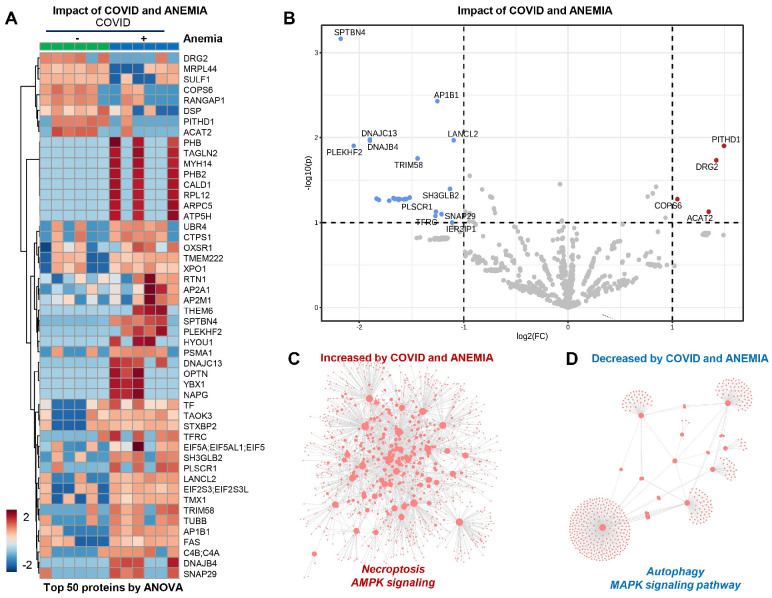
(Patients from Group 2) (**A**,**B**) Effect of anemia appreciated from unsupervised hierarchical cluster analysis (HCA) and volcano plot analyses. (**C**) Elevation in protein components involved in necroptosis and AMPK signaling. (**D**) Decreases in the levels of proteins involved in MAPK signaling and autophagy.

**Figure 6 microorganisms-12-00453-f006:**
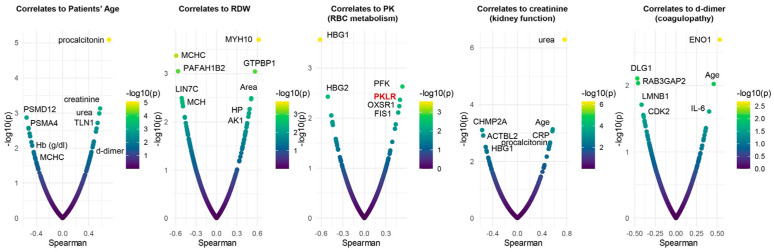
Spearman plot correlations with patient age, RDW, PK activity, kidney function (creatinine), and coagulability (d-dimer). As an internal validation of the quality of the observed correlative results, enzymatic assays of PK activity correlated significantly with the levels of RBC-PK isoforms (PKR) as detected via proteomics.

**Table 1 microorganisms-12-00453-t001:** Patients’ clinical classification.

PATIENTS		GROUPS	NUMBER of CASES
COVID-19+	without ANEMIA	GROUP 1	13 (18%)
COVID-19+	with ANEMIA	GROUP 2	20 (27%)
VIRAL INFECTION	without ANEMIA	GROUP 3	10 (13%)
VIRAL INFECTION	with ANEMIA	GROUP 4	20 (27%)
HEALTHY CONTROLS		GROUP 5	11 (15%)

**Table 2 microorganisms-12-00453-t002:** Hematological and biochemical parameters. Mean (SD).

		GROUP 1 COVID-19 Positive NO ANEMIA (*n* = 13)	GROUP 2 COVID-19 Positive ANEMIA (*n* = 20)	GROUP 3 VIRAL INFECTION NO ANEMIA (*n* = 10)	GROUP 4 VIRAL INFECTION ANEMIA (*n* = 20)	Normal Controls (*n* = 11)	*p* *
**RBCs (×10^12^/L)**		4.61 (0.67)	3.86 (0.72)	4.53 (0.38)	3.80 (0.74)	4.51 (0.48)	
**Hemoglobin (g/L)**	**M**	142.7 (21.4)	121.4 (17.5)	131.1 (8.2)	108.1 (18.4)	139.0 (16.0)	
	**F**	136.0 (13.0)	107.5 (12.0)	124.5 (12.0]	101.5 (20.0)	131.0 (18.0)	
**MCV (fl)**		91.72 (4.68)	88.98 (6.63)	88.09 (5.28)	87.01 (7.51)	88.28 (3.10)	
**MCH (pg)**		30.85 (1.47)	29.20 (2.68)	29.04 (1.75)	29.03 (3.16)	30.24 (1.24)	
**MCHC (g/L)**		338.5 (8.00)	327.7 (8.90)	329.2 (5.30)	329.8 (8.80)	342.50 (7.20)	
**RDW**		13.52 (0.40)	15.7 (2.33)	14.88 (1.35)	15.58 (3.94)	13.25 (0.52)	
**Reticulocytes (%)**		0.87 (0.50)	1.47 (0.90)	1.15 (0.56)	1.51 (1.19)	1.15 (0.55)	0.125
**Retis (×10^9^/L)**		40.45 (23.85)	53.61 (27.38)	52.23 (25.57)	55.32 (38.58)	51.98 (25.16)	0.537
**WBCs (×10^9^/L)**		6.57 (4.44)	8.23 (4.62)	9.55 (3.88)	10.86 (5.68)	6.04 (1.61)	0.002
**Platelets (×10^9^/L)**		170.77 (73.04)	241.05 (147.99)	225.2 (67.65)	259.6 (126.2)	251.91 (62.9)	0.121
**MPV (fl)**		8.39 (1.18)	8.41 (1.15)	8.41 (1.17)	8.51 (1.03)	8.74 (0.85)	0.703
**Fibrinogen (g/L)**		5.90 (1.32)	5.80 (1.42)	5.88 (5.61)	6.50 (0.97)	3.58 (1.58)	0.009
**D-dimer (ng/L)**		1469 (1653.07)	2858 (2548.69)	1375 (1438.96)	1074 (526.26)	294 (149.46)	0.046
**Procalcitonin (ng/L)**		2.80 (4.00)	58.20 (38.70)	5.30 (9.16)	16.70 (25.6)	5.00 (2.5)	0.001
**Creatinine (mg/L)**		8.70 (3.60)	16.9 (11.70)	10.10 (3.40)	12.20 (6.00)	6.90 (1.40)	0.003
**BUN (mg/L)**		403.60 (296.20)	824.70 (567.70)	536.70 (212.10)	505.2 (449.9)	220 (27.80)	0.001
**PCR (mg/L)**		627.50 (47.59)	767.10 (458.20)	586.10 (439.20)	1180 (775.60)	4.80 (2.10)	0.006
**Hepcidin (nM/L)**		44.71 (35.26)	70.93 (91.47)	194.77 (426.99)	212 (299.54)	9.55 (6.32)	0.009
**Serum iron (mg/L)**		664.02 (410.0)	544.40 (381.8)	465.7 (300.2)	213.0 (118.7)	1035(339.5)	0.002
**Serum ferritin (ng/L)**		1550 (522.50)	5375.3 (4052.2)	3681.40 (2839.80)	4520 (4611.4)	825 (741.7)	0.012
**Serum transferrin (mg/L)**		2136 (592.20)	1572 (561.20)	1747.10 (284.70)	2011 (468.20)	2650 (261.5)	0.002
**Transferrin saturation** **Index (TSI)**		24.00 (5.89)	26.67 (17.21)	21.29 (17.59)	7.80 (4.71)	28.5 (10.07)	0.003
**Serum cobalamin** **(Vit B12) (pg/L)**		577.0 (191.09)	908.63 (760.30)	640.29 (439.93)	614.60 (517.0)	486.43 (218.4)	0.971
**Serum folate (ng/L)**		66.00 (45.7)	59.40 (27.8)	69.90 (47.7)	74.7 (42.0)	55.0 (12.40)	0.809
**Serum bilirubin** **(Total) (mg/L)**		5.10 (1.1)	6.20 (4.5)	4.80 (0.19)	5.60 (2.50)	7.00 (2.90)	0.597
**ALT (IU/L)**		19.8 (12.1)	21.67 (10.32)	25.13 (19.14)	24.47 (13.89)	15.67 (7.23)	0.162
**Alkaline phosphatase (IU/L)**		86.25 (25.36)	72.5 (32.53)	69.86 (34.98)	118.69 (87.5)	50.80 (6.91)	0.085

**Table 3 microorganisms-12-00453-t003:** RBC enzyme activity (IU/gHb). Results are given as mean (SD). Since the majority of RBC enzymes do not follow a normal distribution, nonparametric statistical tests have been used (H of Krushal–Wallis).

	GROUP 1 COVID Positive NO ANEMIA (*n* = 13)	GROUP 2 COVID Positive ANEMIA (*n* = 20)	GROUP 3 COVID Negative NO ANEMIA (*n* = 10)	GROUP 4 COVID Negative ANEMIA (*n* = 20)	Reference Values (*n* = 11)	*p* *
Age	62.15 (21.5)	71.8 (14.23)	70.0 (19.50)	67.8 (18.72)	58.20 (15.50)	
Pyruvate kinase (PK)	13.1 (1.66)	14.40 (2.55)	13.10 (1.68)	12.55 (2.13)	11.23 (3.31)	0.047 *
Hexokinase (HK)	1.10 (0.34)	1.25 (0.36)	1.18 (0.16)	1.14 (0.36)	0.94 (0.34)	0.088
Glucose-6-phosphate dehydrogenase (G6PD)	7.48 (1.15)	7.62 (1.47)	8.53 (1.92)	7.29 (1.24)	7.53 (1.12)	0.517
Glucose phosphate isomerase (GPI)	53.22 (11.8)	55.28 (9.11)	52.13 (11.86)	50.81 (12.20)	53.05 (7.15)	0.729
Adenylate kinase (AK)	226.2 (23.7)	258.5 (38.25)	220.9 (37.50)	249.2 (38.24)	236.5 (33.32)	0.049 *
Phosphofructokinase (PFK)	10.45 (2.80)	11.28 (2.59)	9.73 (2.63)	10.53 (1.66)	9.42 (3.16)	0.412
Phosphoglycerate kinase (PGK)	31.57 (4.68)	32.30 (4.30)	31.67 (4.87)	31.62 (4.89)	30.74 (4.79)	0.765
Triosephosphate isomerase (TPI)	2032 (386)	2185 (435)	1975.4 (256)	2051.2 (246)	1897.1 (292)	0.291
Glutathione peroxidase (GPx)	16.3 (4.65)	18.88 (4.99)	17.17 (4.45)	16.66 (3.68)	16.04 (4.89)	0.403
Glutathione reductase (GR)	8.22 (1.82)	8.91 (2.57)	9.08 (2.61)	9.11 (2.74)	8.15 (1.97)	0.941
Reduced glutathione (GSH)	77.89 (14.7)	75.96 (14.27)	73.73 (7.20)	74.96 (18.79)	81.45 (13.04)	0.500
GSH Stability (+APH)	61.59 (17.3)	65.77 (12.50)	59.15 (18.54)	63.40 (25.26)	65.08 (20.24)	0.86 2

* Compared to normal controls.

**Table 4 microorganisms-12-00453-t004:** Osmoscan parameters. Statistical measurement by parametrical test (* *p* < 0.05).

	COVID	ANEMIA	VIRAL INFECTION	O Min (X ± SD)	EI Max (X ± SD)	O Max (X ±SD)	O Hyper (X ± SD)	Area (X ± SD)
Group 1	positive	NO		145 (5.7)	0.612 (0.01)	311 (8.9)	457 (16.7)	162 (7.45)
Group 2	positive	YES		137 (11.2)	0.607 (0.01)	300 (21.9)	464 (14.9) *	168 (7.5) *
Group 3	negative	NO	YES	136 (8.8) *	0.612 (0.01)	297 (22.6)	463 (13.2) *	172 (3.7) *
Group 4	negative	YES	YES	143 (11.1)	0.611 (0.01)	309 (21.1)	466 (13.7) *	168 (6.2) *
Normal Controls				146 (6.6)	0.613 (0.01)	309 (14.1)	449 (15.1)	160 (6.3)

O min: minimal osmolality (RBC osmotic fragility). EI max: maximum elongation index (RBC deformability). O max: osmolality at EI max (300 mOsm/kg). O hyper: at 50% of EI max (RBC hydration maximal osmolality).

## Data Availability

Data are contained within the article.

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
