# Peer review of "SARS-CoV-2 Infection and Anemia—A Focus on RBC Deformability and Membrane Proteomics—Integrated Observational Prospective Study"

_microorganisms, 2024, doi:10.3390/microorganisms12030453_

Round 1

Reviewer 1 Report

Comments and Suggestions for Authors

Abstract : please remake the rows 20-33 after the clasical structure with methods, results, discussions, connclusions. Avoid, when possible, "our study, our results".

1.Introduction: again, please try to avoid personal considerations and discussions when introductive part of the article. This is a part that needs three ideas to be developed: "why, why now and what brings new your research"

2. Material and method: I suggest that the ability of signing an informed consent should not be one of the inclusion criteria, it is just an instrument common to all subjects included. There is an affirmative sentence that, from my point of vue, stands for bias:"The treatment/s of the patients at admission were not specifically recorded." Well, it should have been recorded, especially for women menstruating or for pregnant women , if included, because these are two conditions that can change the anemic aspect. 

PLease refer more precise to the informed consent, with registration number for the approval.

Results: many figures, lots of data included, but I should prefer that discussions include comparative apreciations between literature and the own results. Refering to Figures 5,6,7,and 8, there are more to say and more to compare.

Conclusions: I do not agree with the structure proposed by the authors( numbered list), and only the last one is in fact a conclusion.

Limitations: I did not find any limitations of the study, and I think there are several to be disclosed.

Author Response

Dear rewiever :

Following your comments and suggestions, please find attached herewith the amended manuscript with the indications of the changes made (in red). I full agree with many of these suggestions that will significantly improve the quality of paper presentation 

Reviewer 2 Report

Comments and Suggestions for Authors

The authors provide intriguing data regarding the observations of changes in red cell levels and anemia in patients with COVID.  These observations may provide valuable information regarding the effects of COVID in patients with regard to impact on red cell behavior and function.  

My major concern with the work is the limited number of subjects involved and the fact that there is missing information and only one location from which the data was collected.  It is hard to justify that the results observed in this case are broad and not isolated due to specific characteristics of the patient population involved.

The authors discuss this limitation in their manuscript but do not address how such limitation may alter the conclusions that are being drawn and the relevance of this work in the absence of additional data and observation.  Please address this.

The authors also do some interesting studies with Ektacytometry to look at possible alterations in red cell behavior.  The data shows no significant changes.  How do the authors explain these findings given data and information provided in a publication which they cite?  Specifically, 

Thomas T, Stefanoni D, Dzieciatkowska M, Issaian A, Nemkov T, Hill RC, et al. Evidence of Structural Protein Damage and 569 Membrane Lipid Remodeling in Red Blood Cells from COVID-19 Patients. J Proteome Res. 2020 Nov;19(11):4455–69.

Can the authors provide an explanation as to why the changes reported by Stefanoni et al. do not result in observable changes in Ektacytometry?

Comments on the Quality of English Language

There are several places where the language used in the manuscript could benefit by additional review.  Overall, I thought that the quality was high, but several examples are in the text that are incorrect or incomplete expressions in English.

Author Response

Dear Reviewer,

As highlighted in the manuscript, a significant limitation of our study is the relatively small cohort size and its confinement to a single location. Our hospital boasts a highly specialized Infectious Diseases Department, which served as a centralized diagnostic hub for COVID-19 infections for numerous health care providers across Catalonia. Regrettably, due to constrained financial resources allocated for proteomics, we were compelled to restrict the number of patients included in our study. Additionally, the surge in hospitalizations at the onset of the pandemic imposed substantial organizational challenges, hindering our ability to fully adhere to the initially designed study protocol. Consequently, this resulted in the exclusion of many potential participants from the study.

Given these constraints, it is clear that the results of our study cannot be universally applied. However, some of our findings are sufficiently intriguing to warrant further investigation, potentially with a more extensive patient cohort in the future. I concur with your assessment regarding the difficulty in asserting that the observed results are representative and not merely anomalies specific to the patient population under study. Nonetheless, it is worth noting that such limitations are not uncommon in cohort studies emanating from single institutions.

Regarding the minimal ektacytometric changes observed despite the proteomic findings, I must admit the absence of a definitive explanation. It is conceivable that, at least within our patient group, the protein alterations triggered by their interaction with the virus were insufficient to significantly affect red blood cell deformability.

Your insights are invaluable to us, and we are committed to exploring these areas further in pursuit of a deeper understanding of the implications of our findings. All the amendments made to the manuscript following your comments and suggestions (in red) can be seen in the attached PDF

Dear Reviewer,

As highlighted in the manuscript, a significant limitation of our study is the relatively small cohort size and its confinement to a single location. Our hospital boasts a highly specialized Infectious Diseases Department, which served as a centralized diagnostic hub for COVID-19 infections for numerous health care providers across Catalonia. Regrettably, due to constrained financial resources allocated for proteomics, we were compelled to restrict the number of patients included in our study. Additionally, the surge in hospitalizations at the onset of the pandemic imposed substantial organizational challenges, hindering our ability to fully adhere to the initially designed study protocol. Consequently, this resulted in the exclusion of many potential participants from the study.

Given these constraints, it is clear that the results of our study cannot be universally applied. However, some of our findings are sufficiently intriguing to warrant further investigation, potentially with a more extensive patient cohort in the future. I concur with your assessment regarding the difficulty in asserting that the observed results are representative and not merely anomalies specific to the patient population under study. Nonetheless, it is worth noting that such limitations are not uncommon in cohort studies emanating from single institutions.

Regarding the minimal ektacytometric changes observed despite the proteomic findings, I must admit the absence of a definitive explanation. It is conceivable that, at least within our patient group, the protein alterations triggered by their interaction with the virus were insufficient to significantly affect red blood cell deformability.

Your insights are invaluable to us, and we are committed to exploring these areas further in pursuit of a deeper understanding of the implications of our findings. All the amendments made to the manuscript following your comments and suggestions (in red) can be seen in the attached PDF

Dear Reviewer,

As highlighted in the manuscript, a significant limitation of our study is the relatively small cohort size and its confinement to a single location. Our hospital boasts a highly specialized Infectious Diseases Department, which served as a centralized diagnostic hub for COVID-19 infections for numerous health care providers across Catalonia. Regrettably, due to constrained financial resources allocated for proteomics, we were compelled to restrict the number of patients included in our study. Additionally, the surge in hospitalizations at the onset of the pandemic imposed substantial organizational challenges, hindering our ability to fully adhere to the initially designed study protocol. Consequently, this resulted in the exclusion of many potential participants from the study.

Given these constraints, it is clear that the results of our study cannot be universally applied. However, some of our findings are sufficiently intriguing to warrant further investigation, potentially with a more extensive patient cohort in the future. I concur with your assessment regarding the difficulty in asserting that the observed results are representative and not merely anomalies specific to the patient population under study. Nonetheless, it is worth noting that such limitations are not uncommon in cohort studies emanating from single institutions.

Regarding the minimal ektacytometric changes observed despite the proteomic findings, I must admit the absence of a definitive explanation. It is conceivable that, at least within our patient group, the protein alterations triggered by their interaction with the virus were insufficient to significantly affect red blood cell deformability.

Your insights are invaluable to us, and we are committed to exploring these areas further in pursuit of a deeper understanding of the implications of our findings. All the amendments made to the manuscript following your comments and suggestions (in red) can be seen in the attached PDF

Dear Reviewer,

As highlighted in the manuscript, a significant limitation of our study is the relatively small cohort size and its confinement to a single location. Our hospital boasts a highly specialized Infectious Diseases Department, which served as a centralized diagnostic hub for COVID-19 infections for numerous health care providers across Catalonia. Regrettably, due to constrained financial resources allocated for proteomics, we were compelled to restrict the number of patients included in our study. Additionally, the surge in hospitalizations at the onset of the pandemic imposed substantial organizational challenges, hindering our ability to fully adhere to the initially designed study protocol. Consequently, this resulted in the exclusion of many potential participants from the study.

Given these constraints, it is clear that the results of our study cannot be universally applied. However, some of our findings are sufficiently intriguing to warrant further investigation, potentially with a more extensive patient cohort in the future. I concur with your assessment regarding the difficulty in asserting that the observed results are representative and not merely anomalies specific to the patient population under study. Nonetheless, it is worth noting that such limitations are not uncommon in cohort studies emanating from single institutions.

Regarding the minimal ektacytometric changes observed despite the proteomic findings, I must admit the absence of a definitive explanation. It is conceivable that, at least within our patient group, the protein alterations triggered by their interaction with the virus were insufficient to significantly affect red blood cell deformability.

Your insights are invaluable to us, and we are committed to exploring these areas further in pursuit of a deeper understanding of the implications of our findings. All the amendments made to the manuscript following your comments and suggestions (in red) can be seen in the attached PDF

Dear Reviewer,

As highlighted in the manuscript, a significant limitation of our study is the relatively small cohort size and its confinement to a single location. Our hospital boasts a highly specialized Infectious Diseases Department, which served as a centralized diagnostic hub for COVID-19 infections for numerous health care providers across Catalonia. Regrettably, due to constrained financial resources allocated for proteomics, we were compelled to restrict the number of patients included in our study. Additionally, the surge in hospitalizations at the onset of the pandemic imposed substantial organizational challenges, hindering our ability to fully adhere to the initially designed study protocol. Consequently, this resulted in the exclusion of many potential participants from the study.

Given these constraints, it is clear that the results of our study cannot be universally applied. However, some of our findings are sufficiently intriguing to warrant further investigation, potentially with a more extensive patient cohort in the future. I concur with your assessment regarding the difficulty in asserting that the observed results are representative and not merely anomalies specific to the patient population under study. Nonetheless, it is worth noting that such limitations are not uncommon in cohort studies emanating from single institutions.

Regarding the minimal ektacytometric changes observed despite the proteomic findings, I must admit the absence of a definitive explanation. It is conceivable that, at least within our patient group, the protein alterations triggered by their interaction with the virus were insufficient to significantly affect red blood cell deformability.

Your insights are invaluable to us, and we are committed to exploring these areas further in pursuit of a deeper understanding of the implications of our findings. All the amendments made to the manuscript following your comments and suggestions (in red) can be seen in the attached PDF

Dear Reviewer,

As highlighted in the manuscript, a significant limitation of our study is the relatively small cohort size and its confinement to a single location. Our hospital boasts a highly specialized Infectious Diseases Department, which served as a centralized diagnostic hub for COVID-19 infections for numerous health care providers across Catalonia. Regrettably, due to constrained financial resources allocated for proteomics, we were compelled to restrict the number of patients included in our study. Additionally, the surge in hospitalizations at the onset of the pandemic imposed substantial organizational challenges, hindering our ability to fully adhere to the initially designed study protocol. Consequently, this resulted in the exclusion of many potential participants from the study.

Given these constraints, it is clear that the results of our study cannot be universally applied. However, some of our findings are sufficiently intriguing to warrant further investigation, potentially with a more extensive patient cohort in the future. I concur with your assessment regarding the difficulty in asserting that the observed results are representative and not merely anomalies specific to the patient population under study. Nonetheless, it is worth noting that such limitations are not uncommon in cohort studies emanating from single institutions.

Regarding the minimal ektacytometric changes observed despite the proteomic findings, I must admit the absence of a definitive explanation. It is conceivable that, at least within our patient group, the protein alterations triggered by their interaction with the virus were insufficient to significantly affect red blood cell deformability.

Your insights are invaluable to us, and we are committed to exploring these areas further in pursuit of a deeper understanding of the implications of our findings. All the amendments made to the manuscript following your comments and suggestions (in red) can be seen in the attached PDF

Dear Reviewer,

As highlighted in the manuscript, a significant limitation of our study is the relatively small cohort size and its confinement to a single location. Our hospital boasts a highly specialized Infectious Diseases Department, which served as a centralized diagnostic hub for COVID-19 infections for numerous health care providers across Catalonia. Regrettably, due to constrained financial resources allocated for proteomics, we were compelled to restrict the number of patients included in our study. Additionally, the surge in hospitalizations at the onset of the pandemic imposed substantial organizational challenges, hindering our ability to fully adhere to the initially designed study protocol. Consequently, this resulted in the exclusion of many potential participants from the study.

Given these constraints, it is clear that the results of our study cannot be universally applied. However, some of our findings are sufficiently intriguing to warrant further investigation, potentially with a more extensive patient cohort in the future. I concur with your assessment regarding the difficulty in asserting that the observed results are representative and not merely anomalies specific to the patient population under study. Nonetheless, it is worth noting that such limitations are not uncommon in cohort studies emanating from single institutions.

Regarding the minimal ektacytometric changes observed despite the proteomic findings, I must admit the absence of a definitive explanation. It is conceivable that, at least within our patient group, the protein alterations triggered by their interaction with the virus were insufficient to significantly affect red blood cell deformability.

Your insights are invaluable to us, and we are committed to exploring these areas further in pursuit of a deeper understanding of the implications of our findings. All the amendments made to the manuscript following your comments and suggestions (in red) can be seen in the attached PDF

Reviewer 3 Report

Comments and Suggestions for Authors

As it is well known, coronavirus (COVID-19) infection is characterized primarily by shortness of breath, persistent cough, and fever. Due to the properties and concentration of RBCs in the blood largely determine the health of the gas exchange process, it is necessary to study the effect of this infection on the properties of cells. Previously, changes in structural and functional proteins depending on the severity of the disease were shown in the red blood cells of patients with COVID-19.

D'Alessandro et al. examine the impact of COVID-19 infection on RBC characteristics and demonstrate that SARS-CoV-2 RNA can interfere with RBC hemoglobin stability, enzyme activity, and cell deformability. The results allow the authors to believe their study may contribute to a better understanding of the mechanisms of anemia (provoked by COVID-19 infection) and can provide proteomic biomarkers to predict risk factors for severe disease outcomes and explore more effective treatment methods.

The authors use proteomics to perform a detailed analysis of RBC composition to achieve this research goal. In addition, cell deformability was studied.

The authors conduct a detailed analysis of the results obtained, which are subsequently carefully analyzed, and the possible mechanisms of infection's influence on the properties of RBCs are carefully analyzed.

The manuscript's text is well-readable and contains sufficient relevant references to previously published studies.

So, I recommend the presented manuscript for publication after slight modification.

I recommend that the authors add a paragraph in the "Discussion" section devoted to possible directions for developing the issue under consideration.

Author Response

Dear reviewer: 

Thank you very much for your kind summary of our manuscript, and the friendly comments. Unfortunately this review platform is a little complicate for me to manage and I am very grateful for not having the need to reply to specific comments and suggestions  I have to mention that other reviewers have asked me to made slight modifications to introduction and discussion that may improved the paper's presentation 

Reviewer 4 Report

Comments and Suggestions for Authors  

The paper “SARS-CoV-2 infection and anemia. A focus on RBC deformability and membrane proteomics. Integrated observational prospective study” analyses the potential interaction of SARS-CoV-2 with red blood cells, their structural membrane proteins, and glycolytic enzymes. The Authors performed a wide proteomics investigation, which can help in unravelling SARS-CoV-2 mechanisms underlying anemia. This understanding can be beneficial for potential therapeutic interventions. The conclusions support the results and the information provided is 

clear and well presented. 

Overall, the paper is extremely interesting with data worth of being published. 

Some minor issues: 

Include the details of the Ethical approval (protocol number and date of approval). 

Table 1 is badly edited and confusing. Please modify it. 

Rephrase the conclusions, please show the main results in a more schematic paragraph and not a connection between the phrases, which is difficult to understand. 

There are minor typos throughout the manuscript. 

I am not qualified to judge the statistics and the proteomic analysis.

Author Response

Thank you very much for your kind comments and suggestions.

1.We have modified the Table 1 to present it in a more understanding format .

2.We have added the details of Ethics Approval and Protocol Approval which date is September 30 of 2021(highlighted in yellow)  

Round 2

Reviewer 1 Report

Comments and Suggestions for Authors

I have carefully read the imporved version of the text. Nevertheless, I strongly suggest a global more "scientific writing way" for presenting your work. 

Example:  " Introduction: The effects of COVID-19 are not limited to the respiratory system but also involve complex interactions with other physiological systems. We present here the results of an investigation on the potential link between SARS-CoV-2 infection and anaemia, with a focus on red blood cell (RBC) deformability, haemoglobin stability, enzyme activities, and proteomics."

This is not a way to start the abstract, nor a medical language. Please, try and formulate in a scientific, professional manner your data, as well as insert commas, for a better understanding of the text.

Figure 6 cannot be read in the present form and figure 7 should be erased.

There are still no limitations of the study and Conclusions lack in clarity.

Comments on the Quality of English Language

I have mentioned few of the problems regarding the content and mostly the form of presentation, that have to be corrected, in my opinion, of course.

I suggest a more carefull and scientific, professional way of writing this article, with better care for how the data are presented.

Author Response

Dear Reviewer,

Thank you very much for your extremely valuable feedback, which has significantly improved the presentation of our paper. Should you feel that the revisions made to the manuscript do not yet meet the highest standards, please do not hesitate to inform me.

The amendments are as follows:

  1. I have endeavored to enhance the versions of the Abstract, Introduction, and Discussion sections, aiming for a more scientific style of writing. While I am unsure if these revisions fully reach the desired high standard, I believe they represent a significant improvement in the manuscript's overall articulation.

  2. I have removed Figures 6 and 7 from the document.

  3. Efforts have been made to clarify the conclusions, aiming for greater precision and clarity in our final assertions.

Your guidance is greatly appreciated, and I look forward to any further suggestions you may have.

Best regard 
